# COVID-19 CPR—Impact of Personal Protective Equipment during a Simulated Cardiac Arrest in Times of the COVID-19 Pandemic: A Prospective Comparative Trial

**DOI:** 10.3390/jcm11195881

**Published:** 2022-10-05

**Authors:** Timur Sellmann, Maria Nur, Dietmar Wetzchewald, Heidrun Schwager, Corvin Cleff, Serge C. Thal, Stephan Marsch

**Affiliations:** 1Department of Anaesthesiology and Intensive Care Medicine, Bethesda Hospital, 47053 Duisburg, Germany; 2Department of Anaesthesiology 1, Witten/Herdecke University, 58455 Witten, Germany; 3Institution for Emergency Medicine, 59755 Arnsberg, Germany; 4Department of Anaesthesiology and Intensive Care Medicine, Faculty of Medicine and University Hospital Cologne, University of Cologne, 50937 Cologne, Germany; 5Department of Intensive Care, University Hospital, 4031 Basel, Switzerland

**Keywords:** COVID-19, personal protective equipment, cardiopulmonary resuscitation, simulation, controlled trial

## Abstract

*Background:* Guidelines of cardiopulmonary resuscitation (CPR) recommend the use of personal protective equipment (PPE) during the resuscitation of COVID-19 patients. Data on the effects of PPE on rescuers’ stress level and quality of CPR are sparse and conflicting. This trial investigated the effects of PPE on team performance in simulated cardiac arrests. *Methods:* During the pandemic period, 198 teams (689 participants) performed CPR with PPE in simulated cardiac arrests (PPE group) and were compared with 423 (1451 participants) performing in identical scenarios in the pre-pandemic period (control group). Video recordings were used for data analysis. The primary endpoint was hands-on time. Secondary endpoints included a further performance of CPR and the perceived task load assessed by the NASA task-load index. *Results:* Hands-on times were lower in PPE teams than in the control group (86% (83–89) vs. 90% (87–93); difference 3, 95% CI for difference 3–4, *p* < 0.0001). Moreover, PPE teams made fewer change-overs and delayed defibrillation and administration of drugs. PPE teams perceived higher task loads (57 (44–67) vs. 63 (53–71); difference 6, 95% CI for difference 5–8, *p* < 0.0001) and scored higher in the domains physical and temporal demand, performance, and effort. Leadership allocation had no effect on primary and secondary endpoints. *Conclusions:* Having to wear PPE during CPR is an additional burden in an already demanding task. PPE is associated with an increase in perceived task load, lower hands-on times, fewer change-overs, and delays in defibrillation and the administration of drugs. (German study register number DRKS00023184).

## 1. Introduction

The current European Resuscitation Council COVID-19 guidelines recommend the use of personal protective equipment (PPE) during the cardiopulmonary resuscitation (CPR) of a suspected COVID-19 victim [1,2]. These recommendations are based on considerations of potential droplet and airborne transmission from the victim to the rescuer during CPR. In addition, international and national recommendations regarding a hygienic faultless dressing (“donning”) and removal (“doffing”) of PPE have also been published in order to further minimise the risk of contagion [3]. Data concerning CPR during COVID-19 is still limited and the outcome seems worse in comparison to non-COVID-19 patients [4,5]. Two large studies including over 6300 COVID-19 patients reported no survival [6] and 3% survival in patients aged > 79 years, respectively [7]. The authors of both studies advocate the need for further studies of CPR in COVID-19 and claim that strategies are needed to further optimize CPR for COVID-19, including the use of PPE [6,7]. Current data on the impact of PPE on CPR mostly relate to small studies investigating the quality of chest compressions of single rescuers` in simulated arrests [8,9,10,11,12,13,14,15]. Results are conflicting with studies reporting no effect of PPE [11,12] or negative effects [8,13,14,15,16]. 

The beneficial effects of PPE include counteracting aerosol or droplet transmission-induced infection rates [1,2,3]. Delays by “donning” PPE in COVID-19 CPR has been discussed [2], but published data preferentially did not show any relevant delay during life-saving procedures in various populations [11,17]. CPR is a stressful task in itself [18] and the need to wear PPE may add supplementary cognitive and emotional demands. Finally, there are no data, though important in CPR teams [19,20], on whether leadership is able to mitigate the effects of additional stress, as may be caused by PPE.

Investigating the impact of PPE on the quality of CPR in COVID-19 patients in large randomized trials would be difficult in real cases for a variety of reasons. Simulation allows the investigation of team performance both globally and in specific subtasks in a realistic and standardized manner [21]. A particular advantage of simulation is the possibility of recording data right from the start. Accordingly, the aim of this trial was to investigate the effects of PPE on rescuers’ perceived task load and the quality of CPR, and the mitigating effects of team leadership, if any, in simulated cardiac arrests. 

## 2. Material and Methods

### 2.1. Participants

The Working Group on Intensive Care Medicine, Arnsberg, Germany (http://www.aim-arnsberg.de; assessed on 19 August 2022), organizes educational courses for physicians, mainly residents in their 2nd to 3rd year of postgraduate medical education in internal medicine, anaesthesia, or surgery, from Germany and German-speaking countries working in intensive care and emergency medicine. Participants of the courses were offered to attend voluntary simulator-based CPR workshops and were informed that simulations were video-taped for scientific reasons. Identical workshops were offered to physicians wishing to participate without being filmed. To capture the actual level of training of the course participants, a specific theoretical instruction on their knowledge of CPR was omitted. The updates in international CPR guidelines (AHA in 2020; ERC in 2021) did not affect the primary and secondary outcomes of the present trial. The trial was carried out following the rules of the Declaration of Helsinki and was approved by the Ethics Committee of “Aerztekammer Westfalen-Lippe” (2020-602-f-S) that waived the obligation to obtain consent. The study is registered at the German Clinical Trial Registry (www.drks.de; assessed on 19 August 2022; DRKS-ID: DRKS00023184) and reported herein according to the extensions to the CONSORT statements of the Reporting Guidelines for Health Care Simulation Research [22]. 

### 2.2. Study Design

This is a prospective comparative trial of two cohorts: during the pandemic years 2020 and 2021, all participants of our workshops performed CPR with PPE (PPE group), while a cohort of participants of pre-pandemic workshops of 2016 to 2019 [23,24,25] served as the control group. Apart from the need to wear PPE, the conditions and settings for both cohorts were identical.

Using computer-generated numbers, participants from single workshops were randomly assigned to teams of three to five physicians. Teams were randomly allocated 1:1:1 to no designated leadership (no intervention), designated leadership by the team (team was given the task to designate a leader prior to the start of the scenario), or designated leadership by a tutor (leader was assigned to a randomly chosen team member by the tutor prior to the start of the scenario). The designated leaders wore a coloured scrub cap and could thus be identified on video-recordings. 

### 2.3. Simulator and Scenario 

The mannequin Ambu Man Wireless (Ambu GmbH, Bad Nauheim, Germany) was used. All participants received a standardized introduction to the workshop, the mannequins, and the resuscitation equipment available. Subsequently, all team members were informed that their role during the following scenario was that of a resuscitation team summoned to an unwitnessed cardiac arrest. The victim of the arrest (mannequin) was handed over to the team. Ventricular fibrillation was displayed on the manual defibrillator simulator (ALSI isimulate, iSimulate, LLC, Albany, NY, USA) once the patches were attached. The study period started with the first touch of the patient by one of the participants and ended after the third defibrillation with the return of spontaneous circulation during the following two minutes of CPR. After handing over the victim, tutors, who were instructed to refrain from any intervention until the end of the study period, operated the resuscitation mannequins. The further course of the scenario was at the discretion of the tutor who, after the simulated scenario, gave educational feedback to the teams. 

### 2.4. Personal Protective Equipment (PPE)

The teams were instructed that, like in the real world, they had to put on complete PPE prior to any patient contact. Due to a strict hygiene protocol, N95 masks had to be worn all the time throughout the course. Further PPE consisting of gloves, glasses, gowns, and scrub caps was displayed in sufficient number and different sizes on a table in the scenario room. This made sure that participants were aware that, after “donning”, they had to deal with a medical emergency. The time needed for “donning” was defined as the interval between the first touch of PPE equipment by any team member and the first touch of the patient by any team member.

### 2.5. NASA Task Load Index

Immediately after the completion of their simulation, participants were asked to fill in the NASA task load index (NASA-TLX) questionnaire. The NASA-TLX assesses six domains on visual analogue scales (which range from 0 to 100): mental demand, physical demand, temporal demand, own performance, effort, and frustration [26]. The NASA-TLX has been extensively validated, is easy to administer, and widely used in different domains like flying, driving, teamwork, and medicine [26,27]. Using the unweighted (or “raw”) NASA-TLX, values reach from “0” (minimum) to “100” (maximum), with higher values indicating a higher workload. 

### 2.6. Data Analysis

Data analysis was performed using the video recordings obtained during the simulations. The study period started with the first touch of the patient by one of the participants and ended after the third defibrillation. The first touch of the patient by one of the participants was defined to be the starting point for the timing of all events.

### 2.7. Statistical Analysis

The primary endpoint was the percentage of hands-on time, defined as the time of actual chest compressions divided by the total time interval of the study period. Secondary outcomes included the time of “donning” PPE, adherence to the CPR guidelines, and the NASA-TLX data. The effect of designated leadership was assessed as a secondary outcome for all outcomes analysed.

Data are expressed as medians [IQR], unless otherwise stated. Statistical analysis was performed using SPSS (version 28). Numerical data were analysed by a non-parametric ANOVA, followed by a Mann–Whitney test, if appropriate. The estimates for differences between the medians and their approximate confidence intervals were obtained by the Hodges–Lehmann estimation. Categorical data were analysed using the chi-square test. A *p* < 0.05 (two-tailed) was considered to represent a statistical significance.

## 3. Results

### 3.1. Participants

One hundred and ninety-eight PPE teams (689 participants, 354 male) were compared with 423 control teams (1451 participants). The gender of the leaders was equally distributed between the control groups and the PPE teams (*p* = 0.31). The PPE teams needed 62 (50–78) s for “donning”.

### 3.2. Primary Outcome

The hands-on times were lower in the PPE group than in the control group (86% (83–89) vs. 90% (87–93); difference 3, 95% CI for difference 3–4, *p* < 0.0001). Leadership allocation had no effect on the hands-on times (*p* = 0.77). For more information, please see Figure 1. 

### 3.3. Secondary Outcomes

The performance metrics of CPR are displayed in Table 1. The PPE teams made fewer change-overs and delayed defibrillation and the administration of drugs. Leadership allocation had no effect on any performance marker (Table 2). 

The PPE teams perceived higher task loads than the control group teams (57 (44–67) vs. 63 (53–71); difference 6, 95% CI for difference 5–8, *p* < 0.0001) and scored higher in the domains physical and temporal demand, performance, and effort (Table 3). Leadership allocation had no effect on the overall task load (*p* = 0.63) nor on any of the six domains (Table 2).

## 4. Discussion

This prospective comparative trial demonstrates that wearing PPE while performing CPR increased the rescuers’ task-load and resulted in lower hands-on times, too few change-overs, and delays in defibrillation and the administration of drugs. Designated leadership did not mitigate the additional burden imposed by PPE.

To the best of our knowledge, this is the first large-scale trial on the effects of PPE in CPR teams. Simulator-based cross-over studies from the pre-COVID-19 area reported a significant deterioration in chest compression performance in participants wearing level-C PPE [8,13,14]. Studies conducted during the COVID-19 pandemic show conflicting results: in a simulator-based randomized study involving 36 residents, no significant effect of PPE on both depth and rate of chest compressions, and time for drugs preparation and administration was detected [11]; likewise, a simulator-based randomized cross-over study involving 32 BLS providers reported no negative effect of PPE on the quality of chest compressions [12]; as well as in another randomized controlled non-inferiority triple-crossover study with a total of 48 participants, in which no negative influence of PPE (specifically N95 +/− expiration valve) was found [28]. By contrast, a simulator-based randomized study involving 80 participants (23 physicians, 57 nurses) reported that compared to wearing a surgical mask, wearing an N95 mask increases a rescuer’s fatigue and decreases the chest compression quality during CPR [15]. In simulator-based randomized cross-over studisssses with participants dressed in full PPE during the pandemic, an automated chest compression device proved to be superior over manual chest compressions in both 35 students after the successful completion of an ACLS course [10] and 67 paramedics [9]. Of note, all above mentioned studies assessed the quality of performance of single rescuers only and only one [15] included data on hands-on time, the primary outcome of our trial. 

In keeping with our findings of increased physical demand, performance, and effort in PPE teams, Chen et al. reported that rescuers’ increases in heart rate, mean arterial pressure, and subjective fatigue scores were significantly more obvious with the use of PPE, which was associated with a significant decrease in effective chest compressions [8]. By contrast, Kienbacher et al. were able to demonstrate increased attention when performing basic life support while attention and dexterity were not inferior when wearing PPE, including N95 masks [29]. However, in light of a PPE-related gradual decrease in effective chest compressions resulting from increased rescuers’ fatigue, our finding of too few change-overs in PPE teams is worrisome.

As far as we know, this is the first trial investigating leadership in the context of CPR teams equipped with PPE. Our results indicate that designated leadership is not able to mitigate the additional burden imposed by PPE.

Strengths of this trial include the large sample size and the perfectly identical conditions for all teams. Limitations include the non-randomized design and the absence of real patients. However, randomized trials on PPE in real COVID-19 cases are difficult to conduct. Moreover, simulation is increasingly regarded as an accepted tool for evaluation [21] and performance markers in simulator-based studies show a high agreement with findings in real cases.

Our study population consisted of physicians in their 2nd to 3rd year of residency that, at the time of the study, acted as potential first responders for cardiac arrests in their hospitals. In addition, we refrained from using special teaching, special PPE protocols, or habituation with repetitive exposure prior to testing our participants in the simulated scenario. As such, our results reflect the actual state of our participants’ knowledge and skills and can be extrapolated to real-world settings. Unfortunately, chest compression depth, a benchmark for fatigue during CPR described in previous studies, could not be assessed with our mannequin due to technical limitations.

Clinical studies have established that deviations from CPR algorithms are associated with decreased rates of return of spontaneous circulation and survival to discharge [30,31]. The present experiment observed several shortcomings and deviations from CPR algorithms associated with PPE. While the negative impact of PPE on any single CPR performance marker may be regarded as small, their aggregated effect in combination with the initial delay of CPR due to “donning” may well be of clinical relevance and contribute to poor outcomes of CPR in COVID-19 patients [5,6,7]. 

Our results indicate that training CPR under the conditions which require PPE is warranted and may be a suitable countermeasure against shortcomings and delays. As first responders have to potentially act under various circumstances requiring PPE, PPE sessions should be integrated in regular teaching and training regardless of the current pandemic situation. The participants should be made aware of potential PPE-related increases in rescuers’ fatigue and advised to frequently change-over.

## 5. Conclusions

Having to wear PPE during CPR is an additional burden for rescuers involved in an already demanding task. PPE is associated with an increase in perceived task load and lower hands-on times, fewer change-overs, and delays in defibrillation and the administration of drugs. These findings should be considered in future resuscitation training.

## Figures and Tables

**Figure 1 jcm-11-05881-f001:**
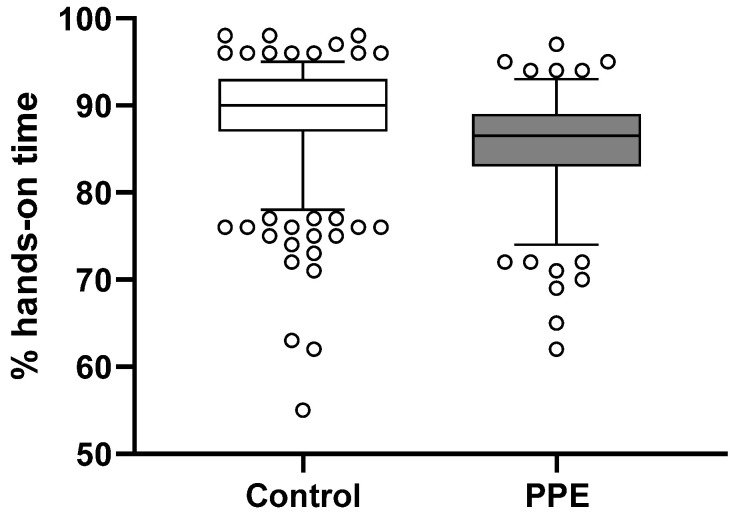
Box and whisker plot of the percentage hands-on time. Boxes represent medians and interquartile range; whiskers delineate the 5th and 95th percentile respectively. Outlier markers indicate teams performing outside the 5th and 95th percentile. White box = control group (no PPE); grey box = group performing CPR with PPE. Hands-on times differed significantly between the groups (*p* < 0.0001).

**Table 1 jcm-11-05881-t001:** Performance metrics of CPR.

	Control Group (*n* = 423)	PPE Group (*n* = 198)	Difference (95% CI)	*p*
Start cardiac massage (s)	12 (8–16)	12 (8–16)	0 (−1–1)	0.49
Chest compression rate (strokes/min)	118 (108–127)	119 (112–124)	1(−3–1)	0.46
Change-overs per 2 min (*n*)	1.3 (1.3–1.7)	0.4 (0.3–0.6)	0.9 (0.9–1.0)	0.0001
AAM completed (CPR cycle)	2 (1–2)	2 (2–2)	0 (0–0)	0.30
Ventilatory rate (breaths/min)	20 (13–28)	18 (12–25)	2 (0–4)	0.28
Time to 1st defibrillation (s)	67 (48–102)	86 (67–119)	19(13–25)	0.001
Time to epinephrine administration (s)	268 (190–312)	319 (282–371)	66 (48–86)	0.001
Time to amiodarone administration (s)	302 (270–340)	423 (388–465)	121 (109–131)	0.0001

Data are medians [IQR]. Estimates for differences between medians and their approximate confidence intervals were obtained by the Hodges–Lehmann estimation. PPE = personal protective equipment; change over = change of the person performing chest compressions (as CPR guidelines recommend a change over every two minutes a value of 1.0 represents perfect adherence); and AAM = advanced airway management.

**Table 2 jcm-11-05881-t002:** Effects of leadership on primary and secondary outcomes.

		No Designated Leadership	Designated Leadership by Team	Designated Leadership by Tutor	*p*
Hands-on time (%)	Control	90 (86–93)	90 (87–92)	90 (87–92)	0.77
	PPE	87 (82–91)	87 (84–89)	85 (82–89)	
Start cardiac massage (s)	Control	13 (7–16)	12 (7–16)	12 (9–16)	0.84
	PPE	12 (8–16)	12 (9–16)	11 (7–14)	
Chest compression rate (strokes/min)	Control	118 (108–129)	118 (106–126)	117 (110–129)	0.38
	PPE	121 (115–121)	119 (112–125)	116 (111–122)	
Change-overs per 2 min (*n*)	Control	1.3 (1.3–1.7)	1.4 (1.2–1.7)	1.3 (1.3–1.7)	0.31
	PPE	0.5 (0.3–0.6)	0.4 (0.3–0.6)	0.4 (0.3–0.6)	
AAM completed (CPR cycle)	Control	2 (2–2)	2 (1–2)	2 (2–3)	0.14
	PPE	2 (1–2)	2 (2–2)	2 (2–2)	
Ventilatory rate (breaths/min)	Control	19 (13–29)	20 (12–29)	20 (15–27)	0.88
	PPE	20 (12–29)	19 (13–27)	17 (9–21)	
Time to 1st defibrillation (s)	Control	66 (50–101)	68 (48–99)	67 (47–107)	0.16
	PPE	86 (65–119)	80 (67–116)	96 (74–121)	
Time to epinephrine administration (s)	Control	264 (201–312)	263 (185–307)	276 (190–317)	0.82
	PPE	324 (283–348)	326 (277–392)	299 (284–373)	
Time to amiodarone administration (s)	Control	298 (262–339)	301 (276–324)	306 (276–359)	0.42
	PPE	418 (380–448)	430 (392–476)	419 (385–465)	
Total task load	Control	56 (46–70)	58 (45–69)	58 (46–68)	0.63
	PPE	63 (56–71)	62 (51–72)	63 (52–69)	
Mental demand	Control	75 (60–85)	70 (55–85)	70 (55–80)	0.30
	PPE	73 (55–85)	70 (50–85)	70 (55–85)	
Physical demand	Control	60 (40–75)	55 (35–75)	60 (35–75)	0.55
	PPE	60 (35–85)	65 (45–75)	60 (30–75)	
Temporal demand	Control	70 (55–85)	70 (50–85)	70 (50–80)	0.41
	PPE	75 (55–85)	70 (50–85)	75 (55–85)	
Performance	Control	60 (35–75)	55 (35–75)	60 (40–80)	0.35
	PPE	68 (45–85)	65 (45–75)	65 (45–75)	
Effort	Control	65 (50–80)	65 (50–80)	65 (50–80)	0.64
	PPE	70 (55–80)	70 (50–80)	65 (50–75)	
Frustration	Control	55 (30–75)	55 (35–75)	55 (30–70)	0.53
	PPE	48 (25–70)	50 (25–70)	50 (35–70)	

Data are medians [IQR] and were analysed using 2-factor ANOVA with group (control or PPE) and leadership allocation as independent between-subject factors. The P-value indicates the effect of the factor leadership. For all outcomes the interactive term group leadership was statistically not significant. PPE = personal protective equipment.

**Table 3 jcm-11-05881-t003:** NASA task load index questionnaire data.

	Control Group (*n* = 1451)	PPE Group (*n* = 689)	Difference (95% CI)	*p*
Total task load	57 (44–67)	63 (53–71)	6 (5–8)	0.0001
Mental demand	70 (50–80)	70 (55–85)	0 (0–5)	0.073
Physical demand	55 (35–75)	60 (35–80)	5 (0–5)	0.005
Temporal demand	65 (50–80)	70 (50–85)	5 (5–10)	0.001
Performance	60 (35–75)	65 (45–80)	5 (5–10)	0.001
Effort	60 (50–75)	65 (50–80)	5 (0–5)	0.001
Frustration	50 (30–70)	50 (25–70)	0 (0–5)	0.87

Data are medians [IQR]. Estimates for differences between medians and their approximate confidence intervals were obtained by the Hodges–Lehmann estimation.

## Data Availability

The data presented in this study are available on request from the corresponding author.

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
