# Peer review of "COVID-19 CPR—Impact of Personal Protective Equipment during a Simulated Cardiac Arrest in Times of the COVID-19 Pandemic: A Prospective Comparative Trial"

_jcm, 2022, doi:10.3390/jcm11195881_

Round 1

Reviewer 1 Report

This is an interesting study on a (still) relevant subject, although the heydays of COVID-19 CPR studies are most likely over.

The main strength of the study is certainly the high number of participants and the per se sound methodology.  There are, however, some issues which in my opinion have to be addressed.

MAJOR:

- This is a study on a CPR scenario, and the outcomes are quality markers of CPR. Yet the authors do not provide any information about the CPR guidelines used. This is especially important because during the study period (2016 to 2021) both the AHA (in 2020) and the ERC (in 2021) published new CPR guidelines. It hence might be possible that not all participants used the same guidelines. Please report in detail which guidelines were taught to the participants and how you dealt with the publication of the guidelines, if this was the case during the study period.

- It is very unfortunate that the participants did not use a manikin which could measure quality of CPR. This is a major limitation in the quality of the measurements. In the methods section there is currently only the information "Data analysis was performed using video recordings". Please provide more detailed information how this was done. Were all measurements done by the same person? Different persons? Different persons in parallel, etc?  This also MUST be added as a limitation in the appropriate section.

- The authors initially aimed to study a potential effect of different types of leadership allocation (none, self, by tutor) on the outcomes, but seemingly found no effect. No data is currently provided in the paper for the different groups (just "no difference"), yet the authors draw conclusions from this comparison in the discussion section and cite it as a strength of their paper. Please provide information on all outcomes for the 3 different groups (for both PPE and non-PPE, respectively), or remove the conclusions on this from the discussion. 

- The authors state that there are currently no RCTs on this topic studying CPR performed by whole teams. This might have been true when the manuscript was originally written, but is not so any more. Please discuss the following papers and similarities/differences with your design and findings:

https://pubmed.ncbi.nlm.nih.gov/33524489/
https://pubmed.ncbi.nlm.nih.gov/35012592/ 

MINOR:

- Please provide a table with information about the participants (at least age and gender, also additional information on level of training, speciality etc. if available) instead of the p-value for gender distribution in line 144

- Please explain the NASA Task load index a little bit more in detail. I am not familiar with this instrument, and other readers might not be as well. What are the minimum/maximum values? 0 and 100? What is better? Higher or lower? What is generally seen as a clinically relevant difference? (i.e. is the difference of 6 points in total task load as found in this study relevant?)

Thanks again for the opportunity to review this paper. I am looking forward to reading the revised version.

Author Response

Reply to Reviewer 1:

Comments and Suggestions for Authors

This is an interesting study on a (still) relevant subject, although the heydays of COVID-19 CPR studies are most likely over.

We thank and agree with the reviewer – still, “after the pandemic is before the pandemic” and we hope that some of the results of our work can be considered for future developments.

The main strength of the study is certainly the high number of participants and the per se sound methodology.  There are, however, some issues which in my opinion have to be addressed.

MAJOR:

- This is a study on a CPR scenario, and the outcomes are quality markers of CPR. Yet the authors do not provide any information about the CPR guidelines used. This is especially important because during the study period (2016 to 2021) both the AHA (in 2020) and the ERC (in 2021) published new CPR guidelines. It hence might be possible that not all participants used the same guidelines. Please report in detail which guidelines were taught to the participants and how you dealt with the publication of the guidelines, if this was the case during the study period.

We thank the reviewer for his valuable comment on guidelines. The updates in international CPR guidelines (AHA in 2020; ERC in 2021) did fortunately not affect the chosen primary and secondary outcomes of the present trial. This information has been added to the methods section of the revised manuscript.

Our research group has been studying the "status quo" of medical resuscitation training in German-speaking countries for some time. Therefore, our research does not include dedicated pre-instruction of specific CPR features. To address this circumstance, we added as follows:

"To capture the actual level of training of course participants, specific theoretical instruction on knowledge of CPR was omitted."

- It is very unfortunate that the participants did not use a manikin which could measure quality of CPR. This is a major limitation in the quality of the measurements. In the methods section there is currently only the information "Data analysis was performed using video recordings". Please provide more detailed information how this was done. Were all measurements done by the same person? Different persons? Different persons in parallel, etc?  This also MUST be added as a limitation in the appropriate section.

The mannequin Ambu Man Wireless theoretically offers the chance to measure CPR quality markers. Due to technical restrictions (camera positioning, blinding participants for “auto-feedback”) a separate evaluation was unfortunately impossible. In order to address this important point, we added as follows in the limitations section (line 222 ff): “Unfortunately, chest compression depth, a benchmark for fatigue during CPR described in previous studies, could not be assessed with our mannequin due to technical limitations.”

- The authors initially aimed to study a potential effect of different types of leadership allocation (none, self, by tutor) on the outcomes, but seemingly found no effect. No data is currently provided in the paper for the different groups (just "no difference"), yet the authors draw conclusions from this comparison in the discussion section and cite it as a strength of their paper. Please provide information on all outcomes for the 3 different groups (for both PPE and non-PPE, respectively), or remove the conclusions on this from the discussion.

We thank the reviewer for his valuable objection. Leadership has always been a secondary outcome parameter in our studies. We believe that an investigation of the influence of leadership in CPR with PPE due to a pandemic has not been studied before and that this alone would be a strength of our study, as it was also described in the discussion. However, in order to provide more insight into our data, we complied with the request of the reviewer and have provided information on all outcomes for the 3 different groups (for both PPE and non-PPE, respectively), in a separate table (new table 3).

- The authors state that there are currently no RCTs on this topic studying CPR performed by whole teams. This might have been true when the manuscript was originally written, but is not so any more. Please discuss the following papers and similarities/differences with your design and findings:

https://pubmed.ncbi.nlm.nih.gov/33524489/

https://pubmed.ncbi.nlm.nih.gov/35012592/

We thank the author for the two valuable related references and added their results into our discussion section.

MINOR:

- Please provide a table with information about the participants (at least age and gender, also additional information on level of training, speciality etc. if available) instead of the p-value for gender distribution in line 144

Unfortunately the only additional information we are able to supply is the number (and with this the percentage) of male participants. We have therefore completed as follows (Results section, line 148): “198 PPE teams (689 participants, 354 male) were compared with 423 control teams (1451 participants).”  Further demographic data as suggested by the reviewer, such as age or level of training or specialty, were not asked for in the course of our investigation. 

- Please explain the NASA Task load index a little bit more in detail. I am not familiar with this instrument, and other readers might not be as well. What are the minimum/maximum values? 0 and 100? What is better? Higher or lower? What is generally seen as a clinically relevant difference? (i.e. is the difference of 6 points in total task load as found in this study relevant?)

The National Aeronautics and Space Administration Task load index (NASA TLX) is a multi-dimensional rating procedure that provides an overall workload score based on a weighted average of ratings on six subscales: Mental, physical and temporal demands as well as own performance, effort, and frustration. The NASA TLX allows to determine the subjective mental workload of a participant while performing a task. The overall workload rating is determined by rating performance as follows:

  1. Mental demand (how much thinking, deciding, or calculating was required to perform the task)
  2. Physical demand (the amount and intensity of physical activity required to complete the task)
  3. Temporal demand (the amount of time pressure involved in completing the task)
  4. Effort (how hard does the participant have to work to maintain their level of performance?)
  5. Performance (the level of success in completing the task)
  6. Frustration level (how insecure, discouraged, or secure or content the participant felt during the task)

Each subscale is presented to the participants either during or after the experimental trial, asking them to rate their score on an interval scale ranging from low (1) to high (20), multiplied x 5 in our study. Initially consisting of nine subscales, the TLX was developed as a paper and pencil questionnaire by NASA Ames Research Center’s Sandra Hart in the 1980s and it has become the gold standard for measuring subjective workload across a wide range of applications since. The actual “pen and pencil” version is accessible for free under https://humansystems.arc.nasa.gov/groups/tlx/downloads/TLXScale.pdf.

The NASA TLX has been successfully used around the world to assess workload in various environments such as aviation (aircraft cockpits); command, control, and communication workstations; supervisory and process control; simulations and laboratory tests and a variety of other domains like healthcare and other complex socio-technical domains. Its use has spread far beyond its original application, focus, and language. So far, it has been translated in over a dozen languages and cited in over 4400 studies highlighting the influence the NASA-TLX has had in human factors research.

In order to correctly assess the importance, it is important to know that listener values mean a higher work load in the respective domain.

The meaning of NASA TLX values are nicely described in Rebecca Grier`s work “How high is high” (Proceedings of the Human Factors and Ergonomics Society 59th Annual Meeting - 2015), where different activities are compared with maximum, unweighted values from daily activities (37.7) to tracking (88.5). In general values > 72 indicate maximum values. Under the assumption, that physicians should also function optimally under stress, our concern was rather to show that differences exist in perceived stress but not in performance. To our knowledge, there are no validated correlations between subjective perceived stress expressed by the NASA TLX and actual performance in the medical field. However, to address the proposed changes, we added the following paragraph in the material and methods section:

“Using the unweighted (or “raw”) NASA TLX, values reach from “0” (minimum) to “100” (maximum), with higher values indication higher workload.” 

Thanks again for the opportunity to review this paper. I am looking forward to reading the revised version.

Thank you very much for the chance to improve.

Reviewer 2 Report

Congratulations to the authors on a well conceived, well designed, well executed, well reported and interesting study. It is unfortunate but not surprising that requiring PPE impedes CPR, but it is good to know that it does and in what respects. This study goes into good detail on the impacts. The detailed assessment carries our understanding beyond the impact of unsurprising delay to include the impacts on CPR quality and provider experience. It is particularly notable (though also regrettable) that leadership cannot mitigate these effects.

No manuscript is perfect. This one is already quite good. I include a marked up copy with some suggestions for minor improvements in case the authors undertake a revision, but they are of insufficient magnitude to require revision.

Author Response

Reply to reviewer 2:

Congratulations to the authors on a well conceived, well designed, well executed, well reported and interesting study. It is unfortunate but not surprising that requiring PPE impedes CPR, but it is good to know that it does and in what respects. This study goes into good detail on the impacts. The detailed assessment carries our understanding beyond the impact of unsurprising delay to include the impacts on CPR quality and provider experience. It is particularly notable (though also regrettable) that leadership cannot mitigate these effects.

No manuscript is perfect. This one is already quite good. I include a marked up copy with some suggestions for minor improvements in case the authors undertake a revision, but they are of insufficient magnitude to require revision.

We would like to thank the reviewer very much for his time and his constructive and consistently positive suggestions. Based on the PDF I have received, I have tried to integrate his suggestions into the manuscript as best as possible. Thank you very much!

Still, we would like to comment on one point addressed in the PDF:

„Here and elsewhere in this table and the next, the IQRs of the compared groups overlap, and even overlap the median of the other group. It is counterintuitive that these are statistically significant differences. Earlier, you have presented the difference with 95% CI of the difference. It could be worthwhile to include  difference with CI statistics in these tables, to raise reader confidence. Some readers will focus on the tables and not see the differences and CIs that are in the text.“

We would like to thank the reviewer for this valuable remark. We have adjusted tables 1 and 2 so that the suggestion is fully met (difference and 95% CI in the tables). We hope that by implementing his suggestion in tables 1 and 2 the justified objection could be completely solved.
